

# Impact of white blood cell count on the development of contrast-induced acute kidney injury in patients receiving percutaneous coronary intervention

Chengxiao Fu[1,2,*], Chenxi Ouyang[3,*], Guoping Yang[1,4], Jingle Li[5], Guiyang Chen[1,5], Yu Cao[5] and Liying Gong[1,6]

[1] Center of Clinical Pharmacology, the Third Xiangya Hospital, Central South University, Changsha, Hunan, China
[2] Department of Pharmacy, The First Affiliated Hospital of University Of South China, Hengyang, Hunan, China
[3] College of Pharmacy, University Of South China, Hengyang, Hunan, China
[4] Center of Clinical Drug Evaluation, the Third Xiangya Hospital, Central South University, Changsha, Hunan, China
[5] Center of Cardiology, the Third Xiangya Hospital, Central South University, Changsha, Hunan, China
[6] Center of Critical Care Medicine, the Third Xiangya Hospital, Central South University, Changsha, Hunan, China
[*] These authors contributed equally to this work.

Corresponding authors
Yu Cao, caoyu0811@csu.edu.cn
Liying Gong, gongliying2016@outlook.com

## ABSTRACT

This study aimed to investigate the efficacy of a pre-procedural white blood cell (WBC) count in the prediction of contrast-induced acute kidney injury (CI-AKI) risk in coronary artery disease patients receiving a percutaneous coronary intervention (PCI). This observational study comprises a sample of 1,013 coronary artery disease patients (including ACS and stable angina) receiving PCI, gathered from September 2015 to July 2017. CI-AKI incidence in the study population was 4.8% (49/1013). Patients in the CI-AKI group had significantly higher WBC counts than those in the non-CI-AKI group ($10.41 \pm 5.37$ *vs.* $8.09 \pm 3.10$, $p = 0.004$). Logistic analysis showed that WBC count (odds ratio [OR]: 1.12, 95% CI [1.03–1.21], $P = 0.006$) was a significant and independent predictor of CI-AKI risk in patients receiving PCI, Receiver-operating characteristic (ROC) curve analysis found that pre-procedural WBC count $\geq 11.03^*10^9$/L was the optimal cut-off value in the prediction of CI-AKI risk with a sensitivity of 41.0% and a specificity of 86.5%. Patients with CI-AKI had a significantly worse 1-year survival rate than patients without CI-AKI (91.8% *vs.* 97.6%, $P = 0.012$). In summary, increased pre-procedural WBC count is associated with an increased risk of developing CI-AKI in patients receiving PCI.

## INTRODUCTION

Contrast-induced acute kidney injury (CI-AKI) is a common complication arising among patients receiving percutaneous coronary intervention (PCI) and is associated with increased morbidity, mortality, and healthcare resource utilization (*Davenport, Perazella & Nallamothu, 2023*; *Davenport et al., 2020*; *Gupta et al., 2005*; *McCullough et al., 2006*; *Rihal et al., 2002*). PCI often requires the administration of an iodinated contrast medium, which is often cited as the cause of CI-AKI. The incidence of CI-AKI is 2% for the general population (*Demircelik et al., 2015*); however, patients receiving PCI are at higher risk of developing CI-AKI, and patients who also suffer from chronic kidney disease and/or diabetes have a risk of almost 50% (*Finn, 2006*; *Goldfarb et al., 2009*).

The prognostic impact of CI-AKI depends on the persistence of renal function deterioration and the degree of kidney injury (*James et al., 2011*; *Maioli et al., 2012*). Currently, there is no effective medication used to treat CI-AKI emphasizes the need for clinical efforts to prevent its development prior to diagnosis (*Toso et al., 2014*). Several conditions and factors, including diabetes, chronic kidney disease, anemia, advanced age, congestive heart failure, and using a large amount of contrast media (CM) in procedures, can increase a patient's chances of developing CI-AKI (*McCullough et al., 1997*; *Mehran et al., 2004*; *Wi et al., 2013*).

The inflammation response has been found to significantly impact the development of CI-AKI (*Demircelik et al., 2015*; *Kocas et al., 2015*; *Kurtul et al., 2016*; *Kwasa, Vinayak & Armstrong, 2014*; *Liu et al., 2012*). White blood cell (WBC) count, leukocyte count, platelet-to-lymphocyte ratio, and WBC differential are widely recognized as indications of inflammation (*Kocas et al., 2015*; *Shah et al., 2017*; *Sun et al., 2018*). *Shah et al. (2017)* discovered an independent association between increased major adverse cardiovascular events (MACE) and baseline WBC count in patients undergoing PCI for acute coronary syndrome (ACS) and stable coronary artery disease. Yet, to our knowledge, no studies have investigated using WBC count as a predictor of CI-AKI risk in PCI patients. Thus, this study aimed to investigate the relationships between pre-procedure WBC count and CI-AKI incidence in patients with coronary artery disease receiving PCI.

## METHODS

The study protocol was approved by the ethics committee of the Third Xiangya Hospital of Central South University, and the approval number was R15006. Each of the subjects enrolled in this study provided informed consent to participate. We have received written informed consent from participants in our study. ClinicalTrials.gov identifier: NCT02888652. This study was performed according to the STROBE checklist.

### Study design and subjects

The present study is a *post hoc* substudy to the research project entitled ''A real-world study of P2Y12 receptor inhibitor therapy in patients with coronary heart disease undergoing PCI''. Briefly, this study was a prospective observation study of consecutive patients who underwent PCI at 2 clinical centers in China between September 1, 2015, and September

1, 2017. Adult patients (18 years of age or older) with documented serum creatinine (SCr) records both pre- and post-PCI (within 72 hr after the procedure) were included. The exclusion criteria for the study included individuals who had failed PCI, those with hemodynamic instability or class IV heart failure (as defined by the NYHA functional classification system), stage 4 or higher chronic kidney disease (CKD), hypersensitivity to contrast medium, anyone who had been exposed to a second contrast medium within one week of the exposure to the first, and those who had received metformin or aminophylline within the 2 weeks prior to PCI.

All patients who were treated with drug-eluting stents and received dual antiplatelet therapy (aspirin associated with clopidogrel or ticagrelor) were recommended for 12 months. At the time of hospital admission, patients received a loading dose of 300 mg aspirin followed by 100 mg/day and either a loading dose of 300 mg clopidogrel followed by 75 mg/day or a loading dose of 180 mg ticagrelor followed by 90 mg/day as P2Y12 inhibitor. PCI was performed according to standard clinical practice and PCI procedures were performed through the radial approach by experienced interventional cardiologists. Professional operators and radiologists can choose to use Visipaque or Iohexol for contrast-enhanced procedures. Procedural anticoagulation was performed by administering unfractionated heparin to target an activated clotting time of 250 to 300 s. At the decision of the professional operator's discretion, perioperative statin (rosuvastatin 20mg/day or atorvastatin 10mg/day), ACEI/ARB inhibitor, beta-blocker, proton pump inhibitor, and intra coronary or intravenous glycoprotein IIb/IIIa inhibitor was also administered during PCI.

ACS was defined as a group of clinical symptoms that can indicate acute myocardial ischemia. This includes ST-segment elevation myocardial infarction (STEMI), non-ST-segment elevation myocardial infarction (NSTEMI), and unstable angina (UA) (*Kumar & Cannon, 2009*); hypertension is diagnosed when systolic blood pressure (SBP) >140 mmHg and/or diastolic blood pressure (DBP) >90 mmHg, or if the individual takes anti-hypertensive medication; low blood pressure (LBP) is defined as blood pressure <90/60 mmHg (*Murat, Kurtul & Yarlioglues, 2015*); diabetes is defined as having a fasting plasma glucose (FPG) level of $\geq 7.0$ mmol/L, or a 2 h oral glucose tolerance test (OGTT) level of $\geq 11.1$ mmol/L, or a history of diabetes, or requiring the use of glucose-lowering agents (*Collaboration NCDRF, 2015*).

All patients were given intravenous hydration. Standard hydration therapy, using saline or bicarbonate solution, was administered at the physician's discretion. All patients received pre- and post-procedural hydration with saline or bicarbonate solution at a rate of one mL/kg/h for 12 h. If a patient's ejection fraction (EF) was below 40% during PCI, hydration was initiated at a rate of 0.5 mL/kg/hr and continued for 18 to 24 h post-emergency PCI (*Levine et al., 2011*; *Stacul et al., 2011*).

**Endpoints and definitions**
CI-AKI was defined as an increase in serum creatinine by $\geq 0.5$ mg/dl or a relative $\geq 25\%$ within 72 h following the procedure without another clear cause for acute kidney injury (*Jarai et al., 2012*; *Leoncini et al., 2014*; *Thomsen, 2003*).

## Samples and data collection

Blood samples were collected at admission and 48–72 h after CM administration to measure Scr concentrations. To define CI-AKI, the highest Scr value after the procedure was considered. Upon hospital admission, a specific clinical laboratory conducted a baseline measurement of the patient's WBC count in the Third Xiangya Hospital of Central South University. Serum glucose levels and lipid profiles were measured while patients were fasting in the morning after administration. The estimated glomerular filtration rate (eGFR) was calculated using the modification of diet in renal disease study equation (MDRD) (*Levey et al., 1999*). All patients underwent a follow-up evaluation at the 1-year mark, either during a clinical visit or *via* telephone contact. An independent clinical events committee, unaware of the details of the study, evaluated all adverse events.

## Statistical analysis

SPSS 25.0 (Armonk, NY, USA) was used to perform statistical calculations. Continuous variables were summarized by means ± SD and median (interquartile range); categorical variables were summarized by percentages. The normality of continuous data was assessed using the Shapiro–Wilks test and the Kolmogorov–Smirnov test. Continuous variables normally distributed were compared using an unpaired student' $t$-test, while non-normal variables were compared using the Mann–Whitney U-test. The association between categorical variables and treatment groups was analyzed using Chi-squared or Fisher exact tests. In determining the factors independently associated with CI-AKI, we considered a comprehensive list of patient characteristics and treatment patterns as candidates. Using forward elimination, a multivariate logistic regression model in which all variables were associated with CI-AKI was developed. During the multivariable model-building process, variables with a $P$ value $\leq 0.15$ and those considered clinically important, biologically plausible, or supported by previously published data were included. Using variance inflation factors (VIF) to detect multicollinearity. The adjusted odds ratios were calculated for each of these variables. ROC curve analysis was utilized to assess the ability of baseline WBC count to differentiate between patients with and without CI-AKI. The optimal cut-off point was determined by identifying the baseline WBC count that maximized the sum of specificity and sensitivity. Kaplan–Meier survival charts were created to determine the difference in 1-year death rate between patients with CI-AKI and those without. To compare survival curves, we used the log-rank test. For all statistical tests carried out, two-tailed $P$ values were used, with statistical significance being defined by a $P$ value of $\leq 0.05$. We predict a sensitivity of 90% and specificity of 90%, while the prevalence rate of CI-AKI was 5%. the required sample size was finally calculated to be 728. The percentage of missing values was less than 1% for all variables in the study. Missing values of categorical variables were imputed to their most common value, and continuous variables to the median of the non-missing values.

## RESULTS

Two centers enrolled 1300 patients for PCI between September 2015 and July 2017. Eleven patients did not complete hydration and were excluded. In total, 1,013 patients met the

study criteria (Fig. 1). In this study, all patients were given intravenous hydration. Of the total patients, 213 (21%) underwent emergency PCI and were treated with sodium bicarbonate for hydration.

All patients completed a follow-up evaluation at the one-year mark, either in person or *via* phone. Table 1 shows the baseline clinical characteristics stratified by CI-AKI, including patient demographics and clinical, biochemical, angiographic, and procedural variables. The incidence of CI-AKI was 4.8% (49/1013). Patients with CI-AKI (referred to as the "CI-AKI group" hereafter) had a significantly higher WBC count than those without CI-AKI (referred to as the "Non-CI-AKI group" hereafter) (10.41 ± 5.37 *vs.* 8.09 ± 3.10 $10^3$ mL, respectively; $P = 0.004$) and had a significantly lower chance of having used statins ($P = 0.022$) compared with patients in the Non-CI-AKI group. Female patients, low blood pressure, peripheral vascular disease (PVD), intra-aortic balloon pump (IABP), heart failure (I-III), lower hemoglobin, and higher fasting glucose were more likely to develop CI-AKI. There were no significant differences in age, hypertension incidence, and type of coronary artery disease between the patients with and without CI-AKI.

Univariate and multivariate logistic regression analyses were used to analyze the effects of multiple variables, as presented in Table 2. Based on the outcomes of the univariate analyses, age ≥75 years, Female, PVD, LBP, ACS, DM, fasting glucose, PVD, eGFR <60 mL/min, heart failure(I-III), WBC count, P/L, Hb, IABP, EF, LDL, HDL, TC, use of Iohexol, and use of statins, were selected for multivariate logistic regression analyses. Apart from low Hb (OR 0.97,95% CI [0.95–0.99]; $P =0.002$), fasting glucose (OR 1.10,95% CI [1.00–1.20]; $P =0.043$), Iohexol (OR 4.55,95% CI [1.37–15.14]; $P =0.014$), WBC count (OR 1.12,95% CI [1.03–1.21]; $P =0.006$) remained significant and independent in predicting the occurrence of CI-AKI. The use of statins was identified as effective in preventing CI-AKI development (OR 0.24,95% CI [0.05–1.21]; $P = 0.044$).

ROC curve analysis of pre-procedural WBC count found that pre-procedural WBC count could be used to predict CI-AKI incidence. [area under the curve = 0.684 (95% confidence interval $0.545-0.723$), $P = 0.002$] (Fig. 2). A cut-off value of $\geq 11.03 \times 10^9$/L was identified as optimal in predicting CI-AKI incidence, providing a sensitivity of 41.0% and specificity of 86.5%.

The follow-up period ended in September 2018, with complete 12-month follow-up data for all patients. One-year mortality occurred in 27 patients (2.7%). The CI-AKI group had a significantly higher incidence of mortality (8.2% *vs.* 2.4% $P = 0.037$). In the multivariate analysis, one-year mortality was associated with CI-AKI (OR 3.47,95% CI [1.12–10.74]; $P =0.031$). The Kaplan–Meier cumulative survival curves of CI-AKI are shown in Fig. 3. Patients who developed CI-AKI had a significantly lower 1-year survival rate than those who did not (91.8% *vs.* 97.6%, $P =0.012$).

## DISCUSSION

The result of our study suggests that an increase in pre-procedure WBC count at the time of treatment may be related to the development of CI-AKI. In addition, the predictive power of pre-procedural WBC count was independent of other known clinical and laboratory-based

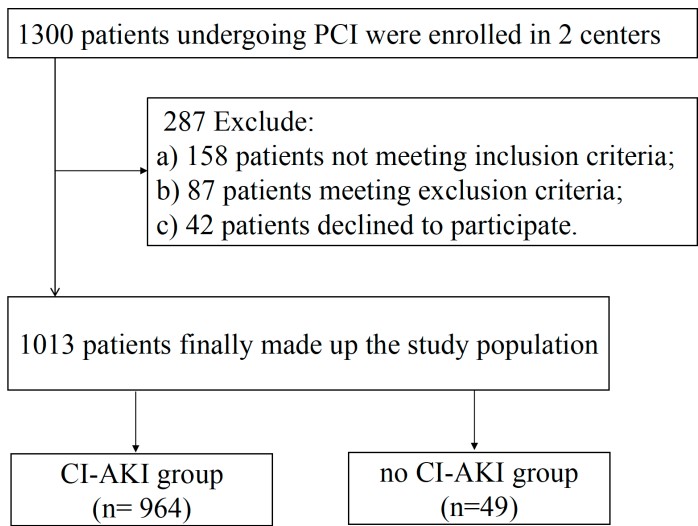

**Figure 1** **Flowchart depicting the inclusion and exclusion of the patients in the present study.** PCI indicates percutaneous coronary intervention; CI-AKI indicates contrast-induced acute kidney injury.

predictors of CI-AKI. A count value of $\geq 11.03*10^9$/L was identified as the optimal cut-off value in the prediction of CI-AKI incidence, with a sensitivity of 41.0% and a specificity of 86.5%.

Some associated conditions and qualities, such as advanced age, anemia, congestive heart failure, diabetes, chronic kidney disease, and large amounts of CM in treatments, have been identified as essential factors predisposing patients to develop CI-AKI (*McCullough et al., 1997*; *Mehran et al., 2004*; *Wi et al., 2013*). Moreover, a number of popularized risk score models, such as the Bartholomew and Mehran scoring systems, have been developed to evaluate a patient's risk of developing CI-AKI (*Bartholomew et al., 2004*; *Mehran et al., 2004*). However, these systems do not consider inflammatory factors and require a patient's entire medical history to assess the risk, making them inconvenient for clinical practice. Clinicians need a sensitive, inexpensive, easily obtainable clinical indicator to assess a patient's risk of CI-AKI prior to PCI.

Thus far, no adjunctive pharmaceutical has been found to effectively treat CI-AKI, which emphasizes the need for accurate clinical assessment methods to prevent CI-AKI development in patients. Evidence from both clinical and experimental studies supports the idea that statins can effectively prevent the development of CI-AKI (*Han et al., 2014*; *Jia et al., 2009*; *Leoncini et al., 2014*). Statins (including rosuvastatin and simvastatin) may be renoprotective *via* several mechanisms, including anti-inflammation, attenuation of endothelial dysfunction, anti-proliferation of mesangial cells, protection of podocytes, and inhibition of the uptake of contrast media into renal tubular cells (*McCullough et al., 2016*). The anti-inflammatory properties of statins are well-documented. Our study also demonstrates that standard dose statins administered in addition to standard hydration can reduce the incidence of renal injury. Statins can reduce the concentrations of pro-inflammatory cytokines and circulating C-reactive protein (CRP) and inhibit vascular

**Table 1  Baseline biochemical and procedural characteristics and medications and lifestyle of the study participants.**

| Variable | Total ($n = 1013$) | Non-CI-AKI Group ($n = 964$) | CI-AKI Group ($n = 49$) | P Value |
|---|---|---|---|---|
| Age, y | $62.70 \pm 10.46$ | $62.67 \pm 10.48$ | $63.39 \pm 10.19$ | 0.637 |
| Female | 284 (28.0%) | 263 (27.3%) | 21 (42.9%) | 0.018 |
| BMI, kg/m$^2$ | $24.31 \pm 3.27$ | $24.33 \pm 3.24$ | $23.93 \pm 3.87$ | 0.415 |
| SBP, mmHg | $132.17 \pm 22.07$ | $132.30 \pm 21.72$ | $129.67 \pm 28.27$ | 0.525 |
| DBP, mmHg | $78.13 \pm 12.52$ | $78.09 \pm 12.29$ | $78.98 \pm 16.55$ | 0.711 |
| **Clinical presentation** | | | | |
| ACS | 873 (86.2%) | 829 (86.0%) | 44 (89.8%) | 0.452 |
| Stable angina | 140 (13.8%) | 135 (14.0%) | 5 (10.2%) | 0.452 |
| **Risk factor** | | | | |
| Age >75 | 101 (10%) | 97 (10.1%) | 4 (8.2%) | 0.810 |
| LBP | 57 (5.6%) | 50 (5.2%) | 7 (14.3%) | 0.016 |
| DM | 464 (45.8%) | 444 (46.1%) | 20 (40.8%) | 0.473 |
| PVD | 379 (37.4%) | 368 (38.2%) | 11 (22.4%) | 0.026 |
| Hb <60, mg/dl | 12 (1.2%) | 12 (1.2%) | 0 (0%) | >0.99 |
| IABP | 15 (1.5%) | 11 (1.1%) | 4 (8.2%) | 0.004 |
| Heart failure (I-III) | 20 (2.0%) | 16 (1.7%) | 4 (8.2%) | 0.013 |
| Kidney transplantation | 0 (0%) | 0 (0%) | 0 (0%) | >0.99 |
| eGFR base line (ml/min/1.73m$^2$) | $74.88 \pm 28.38$ | $74.19 \pm 25.21$ | $88.78 \pm 64.08$ | 0.001 |
| eGFR <60mL/min (ml/min/1.73m$^2$) | 319 (31.5%) | 311 (31.4%) | 8 (33.3%) | 0.844 |
| eGFR after PCI (ml/min/1.73m$^2$) | $80.05 \pm 27.96$ | $81.11 \pm 27.66$ | $58.65 \pm 25.44$ | <0.01 |
| Smokers | | | | |
| Current smoker | 383 (37.8%) | 366 (38.0%) | 17 (34.7%) | 0.645 |
| Former smoker | 169 (16.7%) | 165 (17.1%) | 4 (8.2%) | 0.101 |
| Hypertension | 662 (65.4%) | 629 (65.2%) | 33 (67.3%) | 0.763 |
| TIA/Stroke | 118 (11.6%) | 113 (11.7%) | 5 (10.2%) | 0.747 |
| Previous PCI | 134 (13.2%) | 129 (13.4%) | 5 (10.2%) | 0.522 |
| Previous CABG | 6 (0.6%) | 6 (0.6%) | 0 (0%) | >0.99 |
| **Angiographic data and treatment** | | | | |
| Multivessel disease | 829 (81.8%) | 787 (81.6%) | 42 (85.7%) | 0.571 |
| Multivessel PCI | 657 (64.9%) | 629 (65.2%) | 28 (57.1%) | 0.283 |
| LCA | 67 (6.6%) | 60 (6.2%) | 7 (14.3%) | 0.037 |
| LAD | 945 (93.3%) | 899 (93.3%) | 46 (93.9%) | >0.99 |
| Contrast media dose, mL | $87.97 \pm 9.80$ | $87.96 \pm 9.77$ | $88.27 \pm 10.49$ | 0.830 |
| Iohexol | 735 (72.6%) | 691 (71.7%) | 44 (89.8%) | 0.037 |
| **Laboratory** | | | | |
| WBC,10$^9$/L | $8.21 \pm 3.28$ | $8.09 \pm 3.10$ | $10.41 \pm 5.37$ | 0.004 |
| N,10$^9$/L | $6.30 \pm 5.89$ | $6.24 \pm 5.87$ | $8.74 \pm 6.37$ | 0.027 |
| L,10$^9$/L | $1.70 \pm 1.21$ | $1.70 \pm 1.22$ | $1.46 \pm 0.76$ | 0.255 |
| Hb, g/L | $133.15 \pm 17.27$ | $133.41 \pm 17.16$ | $127.94 \pm 18.55$ | 0.030 |
| Platelet | $212.40 \pm 60.54$ | $211.99 \pm 60.49$ | $220.55 \pm 61.48$ | 0.334 |

| Variable | Total ($n = 1013$) | Non-CI-AKI Group ($n = 964$) | CI-AKI Group ($n = 49$) | P Value |
|---|---|---|---|---|
| P/L | 152.52 ± 83.45 | 151.56 ± 83.58 | 171.20 ± 79.35 | 0.108 |
| LDL cholesterol | 2.34 ± 0.89 | 2.32 ± 0.88 | 2.58 ± 0.99 | 0.056 |
| HDL cholesterol | 1.14 ± 0.28 | 1.14 ± 0.28 | 1.21 ± 0.22 | 0.089 |
| Triglycerides, mmol/L | 1.81 ± 1.43 | 1.82 ± 1.45 | 1.72 ± 1.08 | 0.661 |
| Total cholesterol, mmol/L | 4.47 ± 1.28 | 4.45 ± 1.28 | 4.80 ± 1.24 | 0.072 |
| Fasting glucose, mmol/L | 6.32 ± 2.54 | 6.25 ± 2.38 | 7.66 ± 4.64 | 0.050 |
| Baseline LVEF, % | 58.09 ± 11.91 | 58.35 ± 11.77 | 53.04 ± 13.56 | 0.004 |
| **Medications** | | | | |
| ACEI | 689 (68.0%) | 659 (68.4%) | 30 (61.2%) | 0.296 |
| ARB | 135 (13.3%) | 129 (13.4%) | 6 (12.2%) | 0.819 |
| Beta-blocker | 826 (81.5%) | 786 (81.5%) | 40 (81.6%) | 0.986 |
| Calcium antagonist | 298 (29.4%) | 287 (29.8%) | 11 (22.4%) | 0.272 |
| Statins | 1000 (98.7%) | 954 (99.0%) | 46 (93.9%) | 0.022 |
| GPIIb/IIIa inhibitors | 415 (41.0%) | 391 (40.6%) | 24 (49.0%) | 0.242 |
| Nitrates | 495 (48.9%) | 467 (48.4%) | 28 (57.1%) | 0.235 |
| Proton pump inhibitor | 823 (81.2%) | 779 (80.8%) | 44 (89.8%) | 0.116 |

**Notes.**

Data are expressed as mean ± SD or number of patients or percent (%).

BMI, body mass index; SBP, systolic blood pressure; DBP, diastolic blood pressure; LBP, low blood pressure; DM, diabetes mellitus; PVD, peripheral vascular disease; LCA, left coronary artery; LAD, left anterior descending artery; TIA, transient ischemic attack; ACS, acute coronary syndromes; PCI, percutaneous Coronary Intervention; CABG, coronary artery bypass grafting; WBC, white blood cell; Hb, hemoglobin; N, neutrophil; L, lymphocytes; LDL, low-density lipoprotein cholesterol; HDL, high-density lipoprotein cholesterol; ACEI, angiotensin-converting enzyme inhibitor; ARB, angiotensin receptor blocker; P/L, platelet/lymphocytes; IABP, intra-aortic balloon pump; LVEF, left Ventricular Ejection Fraction.

reactive oxygen species (ROS) generation (*Antoniades et al., 2011*; *Antoniades et al., 2010*; *Ascer et al., 2004*; *Bulcao et al., 2007*; *Van de Ree et al., 2003*). The development of CI-AKI is said to be significantly influenced by inflammation. *Gao et al. (2011)* and *Shacham et al. (2015)* both found that CRP can increase CI-AKI incidence in patients receiving PCI. Test of WBC count, leukocyte count, and WBC differential are widely used as inexpensive indicators of inflammation. Our study supports the idea that these measures of inflammation are useful predictors of CI-AKI risk.

Inflammation is considered an independent risk factor in the development of CI-AKI. One of the most cost-effective markers used in detecting inflammation is WBC count. This study provides additional evidence of the clinical usefulness of a cost-effective and standard measure for potential risk stratification in patients undergoing PCI-related CI-AKI. Emerging evidence suggests that a more robust inflammatory and oxidative response is crucial in the development and progression of CI-AKI. Numerous studies have been conducted to evaluate the link between inflammation and the incidence of CI-AKI (*Akin et al., 2015*; *Gao et al., 2011*; *Kurtul et al., 2016*; *Yuan et al., 2017*). *Yuan et al. (2017)* found an association between various clinical measures (WBC count, N count, and CRP levels) and CI-AKI incidence in ACS patients who undergo PCI. Similarly, *Gao et al. (2011)* also found an association between CRP level and the incidence of CI-AKI in patients who underwent PCI. In accordance with these studies, we found that a higher pre-procedure WBC count was independently associated with the development of CI-AKI. Our findings

**Table 2** Univariate and multivariate analysis of the risk factors related to CI-AKI in patients receiving PCI.

| Variable | Univariate Analysis | | | Multivariate Analysis | | |
|---|---|---|---|---|---|---|
| | OR | 95%CI | P v alue | OR | 95%CI | P value |
| Female | 2.00 | (1.12–3.58) | 0.020 | 1.51 | (0.65–3.54) | 0.342 |
| Age ≥ 75 years | 0.80 | (0.28–2.26) | 0.666 | 0.42 | (0.09–2.01) | 0.277 |
| LBP | 3.05 | (1.31–7.12) | 0.01 | 0.95 | (0.27–3.35) | 0.941 |
| DM | 0.81 | (0.45–1.45) | 0.473 | 0.65 | (0.29–1.44) | 0.289 |
| Heart failure (I-III) | 5.27 | (1.69–16.40) | 0.004 | 1.53 | (0.31–7.56) | 0.602 |
| ACS | 1.43 | (0.56–3.68) | 0.454 | 0.78 | (0.27–2.21) | 0.635 |
| PVD | 0.47 | (0.24–0.93) | 0.030 | 0.70 | (0.32–1.52) | 0.369 |
| eGFR <60 | 0.96 | (0.51–1.79) | 0.892 | 1.08 | (0.49–2.38) | 0.852 |
| Previous smoker | 0.43 | (0.15–1.21) | 0.111 | 0.92 | (0.28–3.01) | 0.892 |
| IABP | 7.70 | (2.36–25.13) | 0.001 | 2.29 | (0.47–11.22) | 0.305 |
| Multivessel PCI | 0.71 | (0.40–1.27) | 0.248 | 0.56 | (0.29–1.10) | 0.092 |
| Multivessel disease | 1.35 | (0.60–3.05) | 0.472 | | | |
| LAD | 1.11 | (0.34–3.66) | 0.866 | | | |
| LCA | 2.51 | (1.08–5.83) | 0.032 | 2.27 | (0.79 –6.56) | 0.129 |
| WBC count | 1.15 | (1.08–1.22) | <0.001 | 1.12 | (1.03–1.21) | 0.006 |
| Hb | 0.98 | (0.97–1.00) | 0.031 | 0.97 | (0.95–0.99) | 0.002 |
| P/L | 1.00 | (1.00–1.01) | 0.110 | 1.00 | (0.99–1.01) | 0.438 |
| Fasting glucose | 1.15 | (1.06–1.25) | 0.001 | 1.10 | (1.00–1.20) | 0.043 |
| Statin | 0.16 | (0.04–0.60) | 0.007 | 0.24 | (0.05–1.21) | 0.044 |
| Contrast media dose | 1.00 | (0.97–1.03) | 0.829 | | | |
| Iohexol | 3.48 | (1.36–8.86) | 0.009 | 4.55 | (1.37–15.14) | 0.014 |
| LDL cholesterol | 1.33 | (0.99–1.78) | 0.056 | 1.97 | (0.86–4.49) | 0.109 |
| HDL cholesterol | 2.38 | (0.88–6.46) | 0.089 | 1.99 | (0.53–7.51) | 0.308 |
| Total cholesterol | 1.20 | (0.98–1.47) | 0.072 | 0.69 | (0.37–1.31) | 0.258 |
| Baseline LVEF, % | 0.97 | (0.94–0.99) | 0.004 | 0.98 | (0.95–1.00) | 0.059 |

**Notes.**
LBP, low blood pressure; DM, diabetes mellitus; PVD, peripheral vascular disease; ACS, acute coronary syndromes; IABP, intra-aortic balloon pump; LCA, left coronary artery; LAD, left anterior descending artery; WBC, white blood cell; P/L, platelet/lymphocytes; PCI, percutaneous coronary intervention; LDL, low-density lipoprotein cholesterol; HDL, high-density lipoprotein cholesterol; LVEF, left Ventricular Ejection Fraction.

align with the results of *Yuan et al. (2017)* but extend to the patients not diagnosed with ACS.

To date, the exact mechanism of CI-AKI development is poorly understood. Iodinated contrast media may cause AKI due to direct toxicity, endothelial dysfunction, oxidative stress, and hemodynamic changes (*Sendeski, 2011*). Although our study cannot fully clarify the plausible mechanism behind the association between higher pre-procedure WBC count and CI-AKI risk, we posit a hypothesis. After administering contrast media during PCI, the highly concentrated contrast media can restrict fluid flow through the medullary vessels and tubules, thus prolonging its contact with the vascular endothelium and the tubular epithelial cells (*Akcay, Nguyen & Edelstein, 2009*; *Seeliger et al., 2012*). As contrast media are commonly cytotoxic, this process causes damage to the tubular and endothelial cells, which as a result, the injured kidneys triggers the transport of various leukocytes, such as

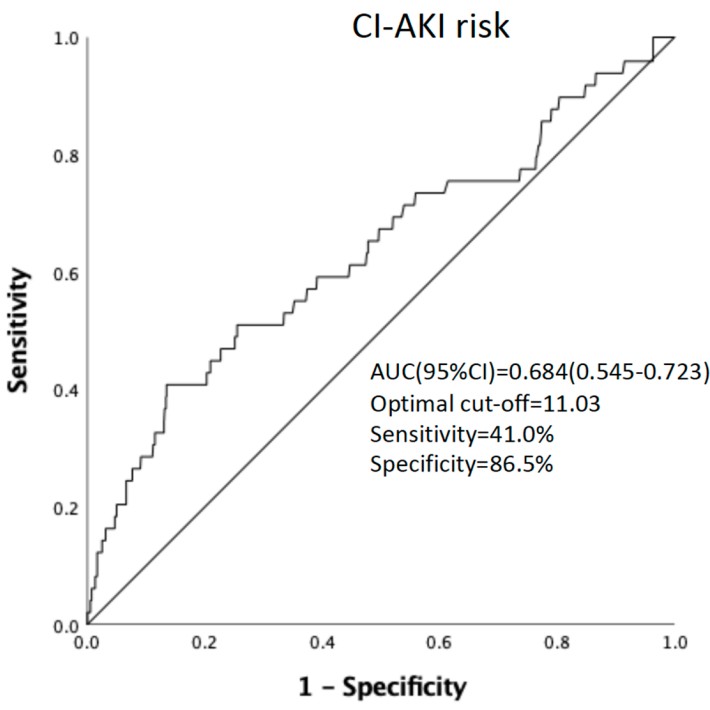

**Figure 2  ROC curve analysis of the pro-procedure WBC count in predicting CI-AKI incidence in PCI patients.**

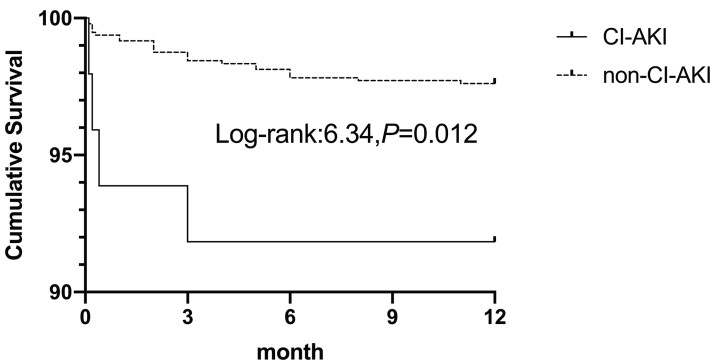

**Figure 3  Kaplan–Meier cumulative survival between patients with CI-AKI and non-CI-AKI.**

neutrophils, lymphocytes, macrophages, and natural killer cells. Such high concentrations of white blood cells can further enhance the generation of inflammatory mediators such as chemokines and cytokines in endothelial cells and tubular cells, contributing to the recruitment of more leukocytes in the kidneys (*Akcay, Nguyen & Edelstein, 2009*). There is a possibility that a higher pre-procedure WBC count combined with contrast media exposure during PCI could significantly increase the risk of CI-AKI.

The development of CI-AKI following PCI can affect both short- and long-term mortality, and morbidity and is an expensive complication (*Caixeta & Mehran, 2010*;

*Liu et al., 2023*; *McCullough et al., 2006*; *Rihal et al., 2002*). Considering the prognostic implications of CI-AKI, the early identification of at-risk patients is extremely important. Emerging strategies such as minimalizing contrast media volume, intravascular volume repletion, intra-aortic balloon pump use, and statins use can significantly reduce CI-AKI risk in patients receiving emergency or elective PCI. ROC curve analysis showed that the baseline WBC count was effective in differentiating between patients who developed CI-AKI and those who did not, however, and so this may not be a significant issue. Using WBC counts to identify patients at higher risk would significantly reduce the number of patients developing CI-AKI and is a technique whose utilization should be strongly considered in clinical practice.

## CONCLUSIONS

Our study confirmed that WBC count is an independent and powerful predictor of risk for CI-AKI in patients undergoing PCI. ROC curve analysis showed that a baseline WBC count of 11.03 was a fair preliminary indicator of creatinine increases, and patients who experienced CI-AKI had a notably lower 1-year survival rate compared to those who did not suffer from CI-AKI. There is mounting evidence supporting the use of WBC count as a prognostic marker for determining CI-AKI risk in patients who receive PCI (*Shah et al., 2017*). To effectively use this marker in clinical settings, further studies are needed to establish treatment options, identify anti-inflammatory medications, and compare its utility with other known inflammatory markers.

## LIMITATIONS

This study is not without limitations. First, it is important to note, however, that this study was merely observational and was conducted in two medical centers with a small sample size which may have affected our results. Second, we excluded patients without measured plasma creatine upon admission or 24–72 h after PCI procedure, which may introduce selection bias. Third, C-reactive protein (CRP) is the important of inflammatory factor, However, we just test CRP levels in some high-risk patients. This study would be more convincing with a detailed analysis of cytokines, for example, TNF-$\alpha$, IL1, and IL6 in the plasma. However, this is just an observation study, so we did not test TNF-$\alpha$, IL1, and IL6 in patients. Finally, Some variables such as age, kidney disease, and diabetes mellitus in previous studies had no statistical significance in our study, which was perhaps because of the strict definition of CI-AKI we used.Patients who developed CI-AKI showed significant differences in their baseline as compared to those who did not. Despite controlling for potential confounding factors related to known comorbidities and drug use, we cannot completely eliminate the possibility of unmeasured confounders affecting our results. Therefore, our findings should be considered as a hypothesis for further confirmation through additional studies.

## ACKNOWLEDGEMENTS

The authors are indebted to the lab staff and the CCU nurses for their precious help in collecting and processing blood samples. They are also very grateful to Manhui Hu and Feng Zheng for their secretarial assistance.

### Funding

This work was supported by the National Natural Science Foundation of China (No. 81803639). The funders had no role in study design, data collection and analysis, decision to publish, or preparation of the manuscript.

### Grant Disclosures

The following grant information was disclosed by the authors:
National Natural Science Foundation of China: 81803639.

### Competing Interests

The authors declare there are no competing interests.

### Author Contributions

- Chengxiao Fu performed the experiments, analyzed the data, authored or reviewed drafts of the article, and approved the final draft.
- Chenxi Ouyang performed the experiments, analyzed the data, authored or reviewed drafts of the article, and approved the final draft.
- Guoping Yang conceived and designed the experiments, prepared figures and/or tables, authored or reviewed drafts of the article, and approved the final draft.
- Jingle Li conceived and designed the experiments, analyzed the data, prepared figures and/or tables, and approved the final draft.
- Guiyang Chen conceived and designed the experiments, prepared figures and/or tables, and approved the final draft.
- Yu Cao conceived and designed the experiments, analyzed the data, prepared figures and/or tables, and approved the final draft.
- Liying Gong conceived and designed the experiments, analyzed the data, authored or reviewed drafts of the article, and approved the final draft.

### Human Ethics

The following information was supplied relating to ethical approvals (*i.e.*, approving body and any reference numbers):

Approval Letter of IRB the Third Xiangya Hospital of Central South University

### Clinical Trial Ethics

The following information was supplied relating to ethical approvals (*i.e.*, approving body and any reference numbers):

The ethics committee of the Third Xiangya Hospital of Central South University

## Clinical Trial Registration

The following information was supplied regarding Clinical Trial registration:

NCT02888652

## Data Availability

The raw measurements are available in the Supplementary File.

## Supplemental Information

Supplemental information for this article can be found online at http://dx.doi.org/10.7717/peerj.17493#supplemental-information.

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
