# Peer review of "Impact of white blood cell count on the development of contrast-induced acute kidney injury in patients receiving percutaneous coronary intervention"

_PeerJ, doi:10.7717/peerj.17493_

## Round 0.1 · original submission · Minor Revisions

Dear authors,

Thank you for your submission and my sincere apologies for the delay in getting a decision. Your manuscript went through peer-reviewing - it is required to go through minor revisions before publication - please, refer to the reviewers' comments for further details. Additionally, please pay attention to the following:

- Include a short background introduction information in your abstract. The language throughout may require a few fine adjustments for which I suggest proofreading for academic English language (and formatting/punctuation incl. spacing), as I cannot accept and send your paper for production until all the fine details are following minimal scientific and communication standards as well as guidelines (for image production, etc). For example, even your abstract would benefit from a few adjustments to improve readability. Please, make sure that all the acronyms have been properly and accordingly defined, incl. in tables and figures. Just a personal suggestion for data visualization: where p is statistically significant, mark it in bold.

**Language Note:** The Academic Editor has identified that the English language must be improved. PeerJ can provide language editing services - please contact us at [email protected] for pricing (be sure to provide your manuscript number and title). Alternatively, you should make your own arrangements to improve the language quality and provide details in your response letter. – PeerJ Staff

Reviewer 1 ·

Basic reporting

The article provides enough context and the referenced literature seems relevant.

Experimental design

The research question is well defined and the methods have been described well. I have the following comments :
1)Did the cohort consist of any patients with chronic inflammatory conditions or recent steroid use that may also cause an elevated white blood cell count.
2)Were ACEi or ARB held prior to the procedure?

Validity of the findings

The results and the conclusion are in alignment. There is no lack of data provided.

Additional comments

A similar study was done: https://onlinelibrary.wiley.com/doi/full/10.1002/clc.22722 . Please remove the statement ‘’ To our knowledge, this is the first prospective study to explore the predictive value of 201 baseline WBC count for CI-AKI development in patients undergoing PCI.’’
However, your study provides data on a 1 year follow up and outcomes in these patients and would like to know if there are any other new information that your study provides that is not currently known?

·

Basic reporting

Dear editor
Thank you for your efforts and help
I would also thank the authors for their astonishing work and brilliant work.
The language was robust, clear and easy to read.
The study design was supportive of the hypothesis. Collecting and analyzing the daya was good.

Experimental design

I have few points though
1.the authors should have reported the percentage of patients who violated the hydration protocol if any.
2.The authors should also report the percentage of patients who recieved bicarbonate or additional hydration as well.

Validity of the findings

3. Mortality at 1 year should adjusted for other known risk factors as well with CI-AKI as one of them

·

Basic reporting

Clear and unambiguous professional English used throughout. Only in the approval letter for the IRB- the English translation seems to be not in synchrony with this study. The clinical trial consent form was not translated in English. Literature references, sufficient field background/context provided.

Experimental design

Original primary research within aims and scope of the journal.
In the design provided in the supplemental data – it seemed to be constructed as a case- control design but it is described as an observational study. The research project entitled “A real-world study of P2Y12 receptor inhibitor therapy in patients with coronary heart disease undergoing PCI” was reviewed and approved by the Institutional Review Board of the Third Xiangya Hospital of Central South University on September 9, 2015- but this was not described in any of the research design. Research question well defined, relevant & meaningful. It is stated how research fills an identified knowledge gap
Rigorous investigation performed to a high technical & ethical standard. Methods described with sufficient detail & information to replicate as above.

Validity of the findings

Conclusions are well stated, linked to original research question & limited to supporting results. More effort to explain why the more proven risk factors for Contrast Induced Nephropathy like age, kidney disease, diabetes mellitus did not show any higher incidence of contrast induced nephropathy in this study.

Additional comments

Was this data collected as part of another study? A real-world study of P2Y12 receptor inhibitor therapy in patients with coronary heart disease undergoing PCI

---

## Round 0.2 · accepted · Accept

Dear authors, I am happy to let you know that your work is being accepted for publication in PeerJ. Please note that now the production office will be in touch with you and careful proofreading should be conducted, thoroughly, in order to avoid typos like the repetition of a sentence around Line 137 and 138.

All the best, Sonia

Reviewer 1 ·

Basic reporting

Changes noted, no comments.

Experimental design

I thank the authors for making the changes. The confounders used to adjust the multivariable analysis seem appropriate. This is my only comment,

- Line 137 and 138 repeated twice – ‘’ Upon hospital admission, a specific clinical laboratory conducted a baseline measurement of the patient's WBC count.”

Validity of the findings

Validity of the findings
The conclusions are in alignment with the methodology.

Conclusion
Well written, no comments

Additional comments

Overall, the changes have made the manuscript better.

·

Basic reporting

The authors answered my points to their best efforts given that the trial was a substudy of another trial

Experimental design

Very good

Validity of the findings

Very good

·

Basic reporting

No comment after corrections

Experimental design

no comment

Validity of the findings

No comment after the explanations

Additional comments

All concerns answered